# Research Progress on the Correlation between the Intestinal Microbiota and Food Allergy

**DOI:** 10.3390/foods11182913

**Published:** 2022-09-19

**Authors:** Hui Yang, Yezhi Qu, Yaran Gao, Shuyuan Sun, Rina Wu, Junrui Wu

**Affiliations:** Key Laboratory of Microbial Fermentation Technology Innovation, Engineering Research Center of Food Fermentation Technology, College of Food Science, Shenyang Agricultural University, Shenyang 110866, China

**Keywords:** food allergy, intestinal microbiota, immune modulation, short-chain fatty acid, probiotic, mechanisms

## Abstract

The increasing incidence of food allergy is becoming a substantial public health concern. Increasing evidence suggests that alterations in the composition of the intestinal microbiota play a part in the development of food allergy. Additionally, the application of probiotics to correct gut microbiota imbalances and regulate food allergy has become a research hotspot. However, the mechanism by which the gut microbiota regulates food allergy and the efficacy of probiotics are still in the preliminary exploration stage, and there are no clear and specific conclusions. The aim of this review is to provide information regarding the immune mechanism underlying food allergy, the correlation between the intestinal microbiota and food allergy, a detailed description of causation, and mechanisms by which the intestinal microbiota regulates food allergy. Subsequently, we highlight how probiotics modulate the gut microbiome–immune axis to alleviate food allergy. This study will contribute to the dovetailing of bacterial therapeutics with immune system in allergic individuals to prevent food allergy and ameliorate food allergy symptoms.

## 1. Introduction

Food allergy is a repeatable atopic disorder resulting from exposure to specific food allergens that can lead to life-threatening allergic reactions. Nowadays, food allergy is becoming increasingly common and represents a serious public health issue which affects approximately 10% of children and 4% of adults worldwide [1]. The allergy often presents as atopic dermatitis, vomiting, diarrhea, and other adverse reactions, potentially leading to anaphylactic shock and death [2]. Therefore, the pathogenesis and treatment of food allergy have attracted more and more attention.

Food allergens are digested and absorbed in the gut, mainly in the small intestine, which contains symbiotic microbiota. With the co-evolution of humans and microbiota, symbiotic microbes inevitably have a considerable impact on human health. Thus, with its high level of microbial richness, the intestinal cavity presents a highly tolerant homeostatic environment in which immune cells respond favorably to food allergens. It has been proven that the human intestinal microbiota, especially the ileum and colon, plays an important role in the intestinal mucosal immunity, by promoting local homeostatic interactions and regulating the immune response to food allergens in the peripheral mucosa [3]. However, the type and number of symbiotic microbiota in the gut may vary and may be affected by the external environment [4]. For example, changes in the dietary patterns, antibiotics use, delivery modes, breastfeeding, vaccines, and pathogen exposure can modify the formation of the intestinal microbiota. Further experimental and epidemiological studies suggest that the gut microbiota composition is related to the clinical trajectory of food allergy and that early life (0–6 month) is a critical period for gut microbial colonization [5]. Several gut microbial metabolites, such as short-chain fatty acids (SCFAs), secondary bile acids (BAs), and amphoteric polysaccharide A (PSA), have been shown to directly or indirectly regulate peripheral induced regulatory T cells (Tregs) differentiation, most of which express the retinoic acid receptor (RAR)-associated orphan receptor γt (RORγt) [6]. RORγt is a master regulator of interleukin (IL)-22-producing immune cells in the gut and can enhance the intestinal epithelial barrier. Furthermore, Feehley et al. (2019) transplanted germ-free mice with feces from healthy or cow-milk-allergic (CMA) infants and found that those mice colonized with healthy infants’ microbiota were protective against allergic reactions to β-lactoglobulin (the major allergen in milk) [7]. Taken together, we can draw a conclusion that there is a strong relationship between the gut microbiota and food allergy, but the exact regulatory mechanism remains unclear. The purpose of this review is to provide information about the immune mechanisms underlying food allergy, the correlation between the gut microbiota and food allergy, and the mechanisms of gut microbiota in regulating food allergy. We also discuss how probiotics can alleviate food allergy by modulating the gut microbiota.

## 2. Food Allergy

According to the different causes, food allergy can generally be divided into two broad categories: immunoglobulin (Ig) E-mediated food allergy and non-IgE-mediated food allergy. In most cases, the immune mechanism of allergic reactions involves an adaptive T helper 2 (Th2)-type response, which subsequently leads to the production of allergens-specific IgE antibodies. The mechanisms of food allergic reaction are shown in Figure 1. The chain of events that occurs in an allergic reaction begins when the allergens are presented to the gastrointestinal immune system, resulting in the production of specific IgE antibodies and damage to the intestinal epithelium. When food allergens pass through the intestinal epithelium again, their infiltration is enhanced. Cytokines such as thymic stromal lymphopoietin, interleukin (IL)-25, and IL-33, which are generated by intestinal epithelial cells, induce OX40L expression in CD103^+^ dendritic cells (DCs) after injury, infection, and immunoactivation, which is beneficial for Th2 cells differentiation. In response, activated Th2 cells release cytokines, including IL-4, IL-5, and IL-13, thus promoting the immunoglobulin class switch of B cells and differentiating into IgE-secreting plasma cells [8].

Through the intestinal epithelium of allergic patients, food allergens are exposed to immune cells mainly located in the mucosa and blood vessels, such as mast cells and basophils, and bind to IgE and its high-affinity immunoglobulin gamma Fc receptor (FcεRI). These interactions lead to the activation of effector T cells and differentiation into Th0 cells, which are stimulated by IL-4 secreted by DCs and further differentiate into Th2 cells. On the surface of mast cells, FcεRI cross-linking triggers a signaling cascade starting from the tyrosine-protein kinase SYK and induces the extracellular secretion of hypersensitive mediators containing histamine, trypsin, and chymosin. These small molecules can induce physiological allergic reactions such as vascular dilatation, increased vascular permeability, pruritus, and synovial contraction, which manifest in the gastrointestinal tract as vascular edema, intestinal congestion, increased intestinal contractility and mucus secretion, acute diarrhea, etc. The factors that result in a deviation in the immune response may include symbiotic microbiota, intestinal epithelial damage, and allergens exposure in other body parts, especially the skin. For instance, many of the receptors that are reported to activate pathogen-associated molecular patterns (PAMPs), including Toll-like receptor 2 (TLR2), TLR5, TLR7, and TLR8, enhance the capacity of DCs to activate Th2 cells [9].

Despite high exposure to food allergens, only a few people experience adverse immunological reactions. The body’s immune failure to respond to food allergens is called immune tolerance. In general, oral tolerance is mainly mediated by CD103^+^ DC_S_, which can present allergens to primary T cells and induce Treg cells to differentiate and proliferate. Meanwhile, CD103^+^DCs can secrete IL-5 and IL-6 to promote the differentiation of B cells into IgA-producing plasma cells [10]. Notably, IgA enhances the barrier function of the intestinal mucosa and maintains the body’s tolerance.

Although genetic factors can cause a predisposition to food allergy, they cannot explain the rapid increase in the prevalence of food allergy; therefore, more and more attention has been paid to environmental factors, such as way of birth, breastfeeding, use of antibiotics, urban or rural settings, intake of junk food or low-fiber, high-fat diets, exposure to pets, etc. All these environmental factors mainly act to regulate the composition and function of gut microbiota. Growing evidence supports the possible role of the gut microbiota in the pathogenesis and progression of food allergy.

## 3. Effect of Intestinal Microbiota on Mucosal Immune System

The human gut contains a wide variety of microbiota, collectively referred to as intestinal microbiota, which includes bacteria, archaea, eukaryotes, and related viruses. *Bacteroidetes* and *Firmicutes* account for more than 90% of the total gut flora, among which obligate anaerobic bacteria is approximately 1000 times the number of aerobic bacteria [11]. The intestinal microbiota of healthy individuals is composed in a certain proportion, and the bacteria are mutually restricted and interdependent to maintain a certain amount and proportion of ecological balance, which plays an important role in the body’s immune response regulation. After birth, newborns are colonized by symbiotic microbiota from the mother’s skin, feces, vagina, and breast milk. *Bifidobacterium* and *Lactobacillus* from the vagina and breast milk are the main genera that colonize the gastrointestinal tract in the first 3 months of life, with *Bifidobacterium* present at the highest abundance [12]. Subsequently, the gut microbial diversity increases after weaning. In adulthood, the main gut bacteria are *Firmicutes*, *Bacteroides*, and *Actinomycetes*. With aging, the abundance of *Firmicutes* and *Bifidobacterium* decreases, while the abundance of *Bacteroides* and *Proteus* increases. The gastrointestinal tract provides a good environment for the growth, metabolism, and reproduction of various facultative and strict anaerobic bacteria [13]. These colonized or transient bacteria rely on nutrients in the gut tract and impact the renewal of gut tissue cells and the immune response [14].

The intestinal mucosal barrier comprises the mucous layer, gut microbiota, and intestinal immune system, and is affected by the integrity of intestinal epithelial cells. The interaction between the intestinal microbiota and the gut barrier maintains relative stability. Notably, the intestinal microbiota establishes the gut immune response and natural defense system, with the latter playing an important role in maintaining the dynamic balance of the gut immune system. Previous studies have confirmed that the microbiota and its community structure affect human health and disease by acting on mucosal barriers. Jakobsson et al. (2015) found that the number of mucus-secreting goblet cells in the cecum of germ-free mice was significantly lower than that of normal mice, and the mucus layer was more unstable [15]. Subsequent studies showed that germ-free mice have abnormal gut morphology and structure, such as decreased total gut area, shortened ileum villi, and a decrease in the number of gut crypts, due to the lack of gut microbiota [16]. Additionally, intestinal microbiota can affect the expression levels of microRNAs in intestinal epithelial cells; however, the relevant mechanism remains unclear. Under normal physiological conditions, the intestinal microbiota promotes the development of the host immune system through specific components such as lipopolysaccharides (LPS), lipoproteins, and metabolites, which form a biological gut barrier [17]. *Lactobacillus acidophilus* mainly parasitizes the gastrointestinal tract of animals, adheres to intestinal epithelial cells, and forms a biological barrier on their surface. Moreover, it reduces the pH of the gut cavity by secreting lactic acid, acetic acid, propionic acid, and other metabolites to prevent the adhesion and reproduction of pathogenic bacteria. Additionally, sodium butyrate (a gut microbial derivative) significantly improved epithelial barrier function of the HT29-MTX-E12 human colonic cell line by increasing mucin 2 (MUC2) levels at concentrations of 1–10 mM. However, sodium butyrate has no positive effect on MUC2 expression at 50–100 mM, probably due to the induction of apoptosis by higher sodium butyrate concentrations [18]. Importantly, ingestion of a butyrate-rich diet increased the expression levels of MUC2 and occludin in the colon, confirming that butyrate facilitates intestinal barrier function [19]. Treatment with butyrate also results in the down-regulation of IL-1β levels, further suggesting that it may help protect against inflammation-induced intestinal barrier damage [20]. Tryptophan metabolites of the intestinal microbiota activate the transcription factor aromatic hydrocarbon receptor (AHR), which promotes immune-cell maturation, and inhibits pathogen colonization in the intestinal tract through IL-22 [21]. The synthesis of AHR ligands oragonists or dietary supplementation of bacterial strains enhances gut barrier function by inducing the secretion of incretin and glucagon-like peptide-1. Certain specialized structures in the gut microbes, such as flagella, pili, and capsules, are often used as antigens to induce an immune response. For example, stimulation of intestinal epithelial cells with *salmonella* flagellin triggers upregulation of CCL20 expression and chemotaxis of immature DCs [22]. Therefore, the intestinal microbiota substantially affects the intestinal mucosal immune system.

## 4. Influence of Dietary Intakes on Intestinal Microbiota Composition

Although colonization and host factors influence the intestinal microbiota composition, dietary intake also plays an important role, and the bacterial populations can be altered with specific dietary interventions.

Studies have shown that a high-fat diet (HFD) alters the composition of the intestinal microbiota. For example, HFD-fed mice showed increased abundances of *Firmicutes*, *Proteobacteria* and *Actinobacteria*, while decreased abundances of *Bacteroidetes* phylum, *Bifidobacterium*, and *Akkermansia* genera compared to normal mice [23]. Meanwhile, HFD consumption in humans resulted in increased abundances of *Blautia*, *Alistipes*, *Bilophila*, several genera of the group *Gammaproteobacteria*, and decreased abundances of *Roseburia*, *Clostridium*, and *Bacteroidess* spp. [24]. Interestingly, gender also influenced the changes in the microbiota after HFD intake with higher abundances of *Campylobacter*, *Blautia*, *Flavonifractor* and *Erysipelatoclostridium*, while the abundances of *Anaerotruncus*, *Eisenbergiella*, *Clostridiales* (FamilyXIIIUCG_001) and *Lachnospiraceae* were higher in males [25]. Protein is a necessary ingredient in dietary and plays an important role in maintaining host health. It is worth noting that a large number of studies have confirmed that dietary protein intake also regulated the diversity of intestinal microbiota, and there are close relationships between different dietary protein sources and intestinal microbiota profiles. As for plant proteins intake, the abundances of *Bifidobacterium* and *Lactobacillus* increased; However, with the intake of animal protein, the abundances of *Bifdobacterium*, *Roseburia*, and *Eubacterium* rectale decreased, and the levels of *Bacteroides*, *Bilophila*, and *Alistipes* markedly increased [26]. Among them, *Roseburia* spp. is the most important and abundant bacteria that are involved in butyrate production in the intestine; *Bifidobacterium* is commonly found in infants’ intestine and increases in these bacteria have been suggested to relieve symptoms of IgE or Th2 allergy [27]. Similarly, animal protein-based diets have been documented to result in increased relative abundances of *Enterococcus*, *Streptococcus*, *Turicibater*, *Escherichia*, *Peptostreptococcaceae*, as well as *Ruminococcaceaea* in mice; in contrast, plant proteins-based diets enriched *Bifidobacteriaceae*, *Desulfovibrionaceae,* and *Coriobacteriaceae* families in mice. Notably, diets based on both animal and plant proteins showed increased abundances of *lactobacilli*, *Lachnospiraceae,* and *Erysipelotrichaceae* [28]. Intake of a fiber-rich diet increased the abundances of *Bifidobacterium*, *Prevotellaceae*, and *Lachnospiraceae* in mice, while decreasing the abundances of *Porphyromonadaceae* and *Lactobacilli* [29]. Vitamins also have the potential to modulate the intestinal microbiota. A study of vitamin A supplementation in mice found that the proportion of *Escherichia-Shigella* could be reduced, while the abundance of *Bacteroides* was significantly increased [30]. Healthy individuals with a higher intake of vitamin D had greater fecal abundance of *Prevotella*, and decreased amount of *Haemophilus* and *Veillonella* [31]. In addition, a recent study by Sun et al. (2018) observed that polyphenols in tea samples could induce the proliferation of certain beneficial bacteria in the intestine and inhibit the growth of harmful bacteria such as *Bacteroides-Prevotella* and *Clostridium histolyticum* [32]. Similarly, a decrease in the amount of *Firmicutes* and *Proteobacteria* phyla was observed in the gut of mice fed cocoa-derived polyphenols, while an increase in the percentage of bacteria belonging to the *Tenericutes* and *Cyanobacteria* phyla [33]. Further studies have shown that dietary polyphenols intake was inversely associated with metabolic syndrome by modulating gut microbiota [34]. Taken together, these results suggest a strong association between specific dietary intakes and intestinal microbiota composition.

## 5. Relationship between the Intestinal Microbiota and Food Allergy

### 5.1. Changes in the Intestinal Microbiota in Patients with Food Allergy

Growing evidence suggests that gut dysbiosis, an imbalance in the intestinal microbiota, plays a decisive role in the development of food allergy. Briefly, the intestinal microbiota composition of people with food allergy differs significantly from that of healthy people, and these differences are more evident in infants and children (Table 1). Christmann et al. (2015) found that children with allergic diseases (including skin, respiratory, and food allergy) exhibited a lower IgG response to specific microbial antigen clusters than healthy children [35]. A study of 166 1-year-old infants with food allergy showed that 7.2% of the patients were sensitive to one or several food allergens, and the intestinal microbiota diversity of allergic infants was significantly lower than that of non-allergic infants [36]. Further studies have shown that the number of specific bacterial groups is linked with the development of food allergy. In particular, the abundance of *Lactobacillus* and *Bifidobacterium* species in 1-week-old infants exhibited a significant negative correlation with the risk of allergy after 5 years [37]. Using 16S rRNA sequencing to compare the fecal microbial composition of allergic and healthy children and found levels of *Clostridium* sensu stricto and *Anaerobacter* in children with food allergy increased and levels of *Bacteroides* and *Clostridium* XVIII decreased [38]. The diversity of total microbiota in children with food allergy was low. Among them, the abundance of *Bacteroidetes* was decreased significantly and the abundance of *Firmicutes* was increased significantly compared with healthy children [39]. Inoue et al. (2017) found that he abundances of *Dorea* and *Akkermansia* were significantly reduced and the abundance of *Veronococcus* was significantly increased in the gastrointestinal tract of infants with food allergy [40]. Lee et al. (2021) speculated that increased abundance of *Ruminococcaceae UCG-002* (Firmicutes phylum), as well as other dominant microbial groups, may reshape the normal gut microbial ecosystem through methane and glycerolipid metabolism pathways into an imbalance state, thereby triggering host IgE-mediated allergic responses [41]. Furthermore, Fazlollahi et al. (2018) investigated the correlation between the intestinal microbiota composition in early life and egg allergy in 141 children and found that the intestinal microbiome diversity and genera of the *Lachnospiraceae* and *Ruminococcaceae* families were related to egg allergy [42]. Bunyavanich et al. (2016) also studied the intestinal microbiota of 226 subjects with milk-allergic subjects and found a significant association between the intestinal microbiota composition, predicted metagenomic function, and remission of milk allergy only in subjects aged 3 to 6 months, whereas *Clostridia* and *Firmicutes* were enriched [43]. This finding has been confirmed in animal experiments. A recent study suggested a correlation between ileal bacteria and genes with upregulated expression levels in the ileum of healthy mice or in the ileum of mice colonized with the CMA infant microbiota, and identified a Clostridium species, *Anaerostipes caccae*, which prevents allergic reactions to food [7]. Decreased *Clostridiales* and increased *Bacteroidales* were also found in the gut of mice with tree nut allergy [44]. It is noteworthy that *Bifidobacterium infantis* and *Bifidobacterium lactis*, which are commonly found in the intestinal tract of nursing mothers and infants, reduce the specific IgE activity in shellfish-sensitized mice [45]. In contrast, other studies have reported no correlation between intestinal microbiota diversity and food allergy [46].

In summary, these findings suggest that the intestinal microbiota is critical for modulating food allergy and suggest that regulating the microbiota community composition may be therapeutically relevant for food allergy. So far, research on the characteristics of the intestinal microbiota in patients with food allergy is still in its infancy, and no specific bacterial taxa has been identified that may be associated with the occurrence of food allergy. Additionally, the main limitation of all these studies is that the number of patients with sensitization or food allergy is small, such that statistical analyses of the effects of potential confounding variables, such as delivery modes, breastfeeding, diet, antibiotic intake, and pets, have not been possible. In addition, several studies have focused on sensitization to food rather than evaluating people with a history of allergic reactions to food or a confirmatory food challenge coupled with skin and/or serum IgE food-specific testing. Moreover, it is recommended to consider the role of the microbiota in food allergy together with the interactions between different taxa and their metabolic effects, rather than just examining bacterial diversity.

### 5.2. Relationship between Intestinal Microbial Metabolites and Food Allergy

The intestinal microbiota is involved in developing and regulating host physiology and immunity by directly participating in the decomposition of dietary components to synthesize and re-synthesize metabolites. Dietary fiber, composed of indigestible carbohydrates extracted from plant polysaccharides and oligosaccharides, is the main source of nutrients for intestinal bacteria, and their fermentation leads to the production of SCFAs (mainly acetic, butyric, and propionic acid). SCFAs are key intermediaries regulating mucosal and systemic immune homeostasis. Intestinal IgA, regulated by the intestinal microbiota, also plays an important role in maintaining intestinal homeostasis and normal function. The SCFAs metabolite, acetate, has been proved to promote intestinal IgA responses mediated by the “metabolite-sensing” receptor, GPR43 [47]. Treatment of mice with the SCFA, propionate, resulted in bone marrow hematopoietic alterations characterized by enhanced generation of macrophages and DC precursors via GPR41, which inhibits Th2 effector cells, thus alleviating allergic inflammation [48]. Moreover, GPR109a is required for butyrate-mediated Treg cells induction and IL-18 in the formation of immune tolerance to food allergens [49]. In addition, SCFAs may also be used as histone deacetylase (HDAC) inhibitors, to increase the acetylation of S6 kinase (S6K), and the phosphorylation of ribosomal protein S6, to regulate the mammalian target of rapamycin (mTOR) pathway, thereby promoting T cells to Th1, Th17, and Treg cell differentiation [50]. Valerate intake increases mTOR activity and production of the anti-inflammatory factor IL-10 [51]. Other studies have shown that acetate increases acetylation of the *Foxp3* promoter by inhibiting HDAC9 and accelerates the production of Treg cells to alleviate allergic diseases [52]. Moreover, SCFAs, combined with G-protein coupled receptors (GPRs), can act on intestinal epithelial cells and activate NLRP3 inflammatory bodies to produce IL-18 [53]. Notably, butyric acid induces protective mechanisms, including the production of mucins and antimicrobial peptides, and increased expression levels of tight junction proteins that strengthen the intestinal epithelial barrier [54].

Emerging evidence has shown that microbiota-derived tryptophan and bile-acid metabolites also play vital roles in food allergy. Tryptophan is an essential amino acid that is degraded to serotonin, kynurenine, or indole in the gut. Supplementation of D-tryptophan produced by *Bifidobacterium* and *Lactobacillus* has been shown to inhibit allergic inflammation in the lungs by increasing intestinal microbial diversity and promoting Treg cell production [55]. Another study suggested that the occurrence of food allergy is associated with an increase in serum indolepropionic acid concentrations, while the presence of multiple food allergies is characterized by a decrease in serum kynurenine and serotonin concentrations [56]. Furthermore, indole 3-propionic acid can effectively modulate the intestinal mucosal system in mice [57]. Secondary BAs produced by intestinal bacteria have also been confirmed to play a key role in regulating intestinal mucosal tolerance by inducing Treg cell differentiation through suppression of the immunostimulatory properties of DCs. Other studies have shown that derivatives of secondary BAs are involved in regulating specific aspects of T cell responses, for example, the reduction in the number of RORγt^+^ Treg cells observed in food-allergic subjects is the result of insufficient microbiota production of secondary BAs metabolites, while these secondary BAs metabolites are associated with RORγt^+^ Treg cell differentiation [58]. Moreover, PSA promotes cytokine production in T cells by activating TLR expression on the surface of DCs and increasing cytokine production by directly binding to receptors on Tregs, thereby maintaining gut homeostasis and health [59].

To sum up, these findings suggest that SCFAs modulate food allergy by activating GPRs, inhibiting HDAC activity, as well as strengthening the protection of the intestinal epithelium barrier (Table 2).

### 5.3. Potential Mechanisms of Intestinal Microbiota in Regulating Food Allergy

Although the occurrence of food allergy is related to the composition of the intestinal microbiota, the exact mechanism is not clear. In this section, we describe in detail the possible mechanisms by which the intestinal microbiota modulates food allergy (Figure 2): (1) regulation of allergen uptake and (2) regulation of signaling molecules involved in immune responses.

#### 5.3.1. Regulation of Allergen Uptake

The intestine contains hundreds of bacterial and various food allergens. Intestinal epithelial cells are physical barriers and primary sites for interactions between the body and the microbiota. Increasing evidence suggests that interactions between the intestinal microbiota and intestinal epithelial cells (mainly microfold cells, goblet cells, and Paneth cells) regulate allergen uptake and maintain the mucosal immune balance. Briefly, the gut microbiota directly interacts with intestinal epithelial cells, promotes the production of defensins, and strengthens the gut barrier function. For example, microfold cells present allergens to DCs and other antigen-presenting cells when stimulated by microbial signals [60]. In addition, the microbiota regulates allergen uptake by inducing Paneth cells to produce antimicrobial peptides and promoting goblet cells to secrete mucus [61]. It is noteworthy that CX_3_CR1^+^ cells, a suspected macrophage cell type, are generally located in the lamina propria of the intestine and are mainly involved in the uptake of allergens from the intestinal lumen. However, when the structure of the intestinal microbiota changes, these cells migrate to other sites, thereby reducing the uptake of allergens. Meanwhile, the intestinal microbiota can interact with DCs, innate immune cells, and macrophages to indirectly enhance the intestinal barrier function and regulate the uptake of food allergens through the production of IL-22.

#### 5.3.2. Regulation of Signaling Molecules Involved in Immune Responses

Intestinal microbiota metabolites interact with host and microbiota, transmit signals to the host, and regulate the immune response. Previous studies showed that SCFAs act on DCs through GPRs and promote Th0 cells to differentiate into Foxp3^+^ Treg cells. Microarray analysis of mouse intestinal epithelial cells revealed that *Clostridia* species modulate innate lymphoid cell function and intestinal epithelial permeability to prevent allergen sensitization [62]. Foxp3^+^ Tregs are one of the major components involved in the body’s immune tolerance system, which can inhibit inflammatory responses and maintain oral tolerance through IL-10, IL-35, and TGF-β [63]. Moreover, Foxp3^+^ Treg cells also promote B cells to secrete more IgA/IgG4 into the gut lumen, thereby inhibiting Th2 cells, decreasing the concentration of Th2-type cytokines and antigen-specific IgE [64].

Recent studies of pattern recognition receptors (PRRs), such as toll-like receptors (TLRs), have elucidated the mechanism of action of the microbiota in the immune regulation of food allergy. PRRs are important part of the innate immune system of the body and exists in various forms. It is not only expressed on the cell membrane, but also widely distributed in the endosomal membrane, lysosomal membrane and cytoplasm. Their role is to monitor the presence of viral molecules, initiate the body’s inflammatory response and antiviral immune signaling pathways, and protect the host from infection. Once PRRs are activated, intestinal epithelial cells produce antimicrobial peptides and mucus to eliminate pathogens and maintain intestinal barrier function. Conversely, the TLR signaling pathway activates gut CD103^+^ DCs, CX_3_CR1^+^ macrophages, and RORt^+^ innate immune cells to secrete cytokines, promote the differentiation of Tregs, and protect against allergy [60]. A lack of TLR-mediated signaling molecules, such as TNF receptor-associated factor 6 (TRAF6), a critical downstream molecule of the TLR signaling pathway, that promotes differentiation of Th2 cells and impaired Treg cells transformation in response to allergens [65]. MyD88 is a downstream molecule involved in TLR signal transduction and inhibits food allergy through the MyD88/RORt+ pathway of regulatory T cells. However, a recent study confirmed the vital role of AHR in regulating and responding to the microbiota. One of the most prominent roles of AHR is the regulation of IL-22 expression. For example, in the absence of AHR, only small amounts IL-22 are produced, immune responses to gut pathogens are reduced, and loss of AHR signaling leads to a disbalance of T cell subsets in the gut, which subsequently changes the composition of the microbiota [66]. Likewise, other studies have demonstrated that AHR activation decreases the number of effector T cells and activates the CD103^+^ DC population to promote oral tolerance to food allergens [67].

Collectively, these findings indicate that various signals from the microbiota must be integrated by various cell subsets in the intestine to alter the balance between sensitization and oral tolerance. When this balance is disrupted, sensitization occurs. TLR and AHR signaling can be received by intestinal epithelial cells, DCs, macrophages, and innate lymphoid cells, enhancing the epithelial barrier, promoting Foxp3+ Treg cell induction, and preventing the generation of food-specific Th2 effector populations.

## 6. Research Progress on the Role of Probiotics in Relieving Allergy

Probiotics are active microbiota that positively impact human health when administered appropriately. They regulate the microbiological balance, inhibit the growth of harmful bacteria, lower blood lipids and cholesterol concentrations, maintain normal immunity of the gut mucosa, and regulate blood pressure [68]. Probiotics mainly include lactic acid bacteria, *Bifidobacteria*, and yeast, with lactic acid bacteria being the primary probiotic bacteria. The probiotics with the greatest potential in promoting the development of the immune system are those belonging to the genus *Lactobacillus* and *Bifidobacterium*. At present, the use of probiotics for the prevention and treatment of allergic diseases mainly focuses on maintaining the Th1/Th2 cell balance, improving intestinal barrier function, and maintaining the balance of intestinal microbiota (Table 3). Reportedly, the addition of *Lacticaseibacillus rhamnosus*, but not *Lacticaseibacillus*
*casei* or *Bifidobacterium*, has been reported to be effective in accelerating the induction of oral tolerance in individuals with milk allergy [69]. Other studies have shown that supplementation with *Lacticaseibacillus rhamnosus* promote the abundance of butyrate-producing strains such as *Lachnospiraceae* and *Ruminoccaceae*, suggesting that probiotics can increase the abundances of tolerance-promoting microbes and promote immune tolerance to food allergens [70]. 

Cell models provide the basis for animal and clinical experiments on probiotics. Many probiotics have been shown to have potential immunomodulatory effects in immune cell models. Although in vitro cell models have some limitations, they have played a crucial role in the preliminary screening of the effects of different bacteria and metabolic components on the immune response. Cell models used to assess the immunomodulatory effects of probiotics include macrophages, human peripheral blood mononuclear cells (PBMCs), mouse spleen lymphocytes, and gut-associated lymphoid tissue. López et al. (2010) tested the specific immune activation properties of different *Bifidobacterium* strains (*B. longum*, *B. breve*, *B. bifidum*, and *B. animalis* subsp) co-incubated with PBMCs in vitro and found that all strains tested induced full DC maturation. However, there were differences in cytokine production levels, especially that of IL-12, IL-10, TNFα, and IL-1β. Probiotic bacteria, cytoplasm, surface extract, and fermentation supernatant also showed immunomodulatory activity, promoting CD8^+^ T cell activation-induced Treg differentiation [83]. In addition to immune cells, certain intestinal epithelial cells, such as Caco-2, T84, and HT-29 cells, have been widely used to study the immunomodulatory effects of probiotics. Caco-2 and HT-29 cells differ from specialized immune cells in that they are cell types that are directly exposed to the gut environment and play important roles in initiating interactions between probiotics and the host. Price et al. (2014) observed that peanut allergens altered intestinal barrier permeability and the tight junction localization in Caco-2 cell cultures [84]. Chichlowski et al. (2012) provided evidence for a specific relationship between human milk oligosaccharide-grown *Bifidobacteria* and intestinal epithelial cells through studies of Caco-2 and HT-29 cells [85]. Moreover, PBMCs from food allergic individuals showed decreased T84 cell barrier integrity compared to those from healthy controls [86].

Fermentation can improve the nutritional value of food by increasing the bioavailability of nutrients and reducing the amount of antinutritional factors. Moreover, fermented food can be used as a functional ingredient with high protein digestibility and probiotics. Several studies have shown that fermentation with probiotics may reduce the immunoreactivity of food allergens. Bu et al. (2010) found that *Lactobacillus helveticus* and *Streptococcus thermophilus* was the most effective combination for reducing the antigenicity of α-lactalbumin and β-lactoglobulin [71]. Other studies have shown that *Lactobacillus*-fermented milk interrupts allergen epitopes, produces bioactive peptides, and regulates the immune system [72]. In addition, lactic acid bacteria fermentation is more conducive to degrade IgE epitopes in wheat allergenic proteins [73]. Consistent with this finding, De Angelis et al. (2007) demonstrated that the probiotic, VSL#3, decreases the number of sensitizing proteins that cannot be degraded by pepsin and trypsin and reduces the potential allergen risk of wheat protein to a certain extent [74].

Many animal experiments and clinical studies have shown that probiotics can be used as an effective tool for allergy prevention and alleviation. *Bifidobacterium* species have been shown to prevent and treat egg allergic reactions by promoting the synthesis of Tregs and inhibiting Th2 cells, to increase the number of CD103^+^ DCs in gut-associated lymphoid tissue CD103^+^ DCs [75]. Fu et al. (2017) found that an oral probiotic, *Bifidobacterium infantis* 14.518 (Binf), effectively suppresses shrimp tropomyosin-induced anaphylaxis in mice via both preventive and therapeutic strategies [76]. Briefly, Binf was found to stimulate DCs maturation and accumulation of CD103^+^ tolerogenic DCs in gut-associated lymphoid tissues, which subsequently induced the differentiation of regulatory T cells and inhibited Th2-biased responses. Neau et al. (2016) identified 3 probiotics with anti-allergic properties, from a total of 31 strains, using PBMCs and Th2-skewed murine splenocytes, and these 3 strains showed a protective effect against sensitization, with decreased concentrations of β-lactoglobulin-specific IgE, IgG1, IgG2a, and mast cell protease [77]. Kozkova et al. (2014) used a mixed gavage of three *Lactobacillus* strains to enhance the gut barrier function; increase sIgA and TGF-β concentrations; reduce gut permeability and decrease the serum concentrations of Bet v 1-specific IgE, IgG1, IgG2a, and alleviate allergic reactions in mice allergic to birch pollen [78]. In addition, *Lacticaseibacillus rhamnosus* GG remarkedly improves the ovalbumin induced food allergy symptoms, most likely by reducing the ratio of IL-4/IFN-γ [79]. The probiotic, Dahi (*Lactococcus lactis* ssp. *cremoris* NCDC-86 and *Lactococcus lactis* ssp. *lactis biovar diacetylactis* NCDC-60), has been shown to skew the Th2-specific immune response toward Th1 and decrease IgE concentration in a whey protein allergy mouse model [80]. Another study showed that *Bifidobacterium brevis* M-16 V alters the intestinal microbiota via IL-33/ST2 signaling pathway and relieves allergic symptoms [81]. Notably, new studies have confirmed that intragastric administration of a probiotic strain, *Lacticaseibacillus casei* BL23, induces Foxp^3+^ and RORγt^+^ Tregs in mice [82]. A similar phenomenon was found in OVA-LPS-induced of allergic asthma inflammation [87]. Obviously, experimental animals are very different from humans in key aspects, such as the gut microbiota composition, immune function or metabolism. Thus, extrapolating the results obtained from animal models to humans may be ineffective and should always be considered as preliminary or limited.

In a clinical study, Capurso (2019) found that administration of *Lacticaseibacillus*
*rhamnosus* GG or a *Bifidobacterium* complete hydrolysis formula significantly improves the condition of skin and ameliorates inflammatory indicators in infants with atopic eczema [88]. Jerzynska et al. (2016) confirmed clinical and immunologic effects of probiotic and vitamin D supplementation for sublingual immunotherapy, for example, in children with allergic rhinitis, where probiotic supplementation demonstrated better clinical and immunotherapeutic effects [89]. Birch pollen allergy has been associated with changes in the fecal microbiota composition, and a probiotic combination of *Lactobacillus acidophilus* NCFMTM (ATCC 700396) and *Bifidobacterium lactis* Bl-04 (ATCC SD5219) has been shown to prevent the pollen-induced eosinophil infiltration into the nasal mucosa and promote a tendency to reduce nasal symptoms [90]. However, the long-term efficacy of probiotics in the prevention and treatment of allergic disease needs to be further evaluated. Plummer et al. (2020) found that postnatal administration of probiotics had no effect on the incidence of allergic disease or atopic sensitization in preterm children during the first 2 years [91]. Moreover, West et al. (2013) reported that *Lactobacillus paracasei* ssp *paracasei* F19 had no effect on any diagnosed allergic disease, airway inflammation, or IgE sensitization [92]. It is worth noting that the results vary from study to study due to multiple factors (dose, probiotics supplement, and environment).

Taken together, many mouse models and human studies have confirmed that single probiotics, mixed probiotics, or some probiotic components have specific preventive and therapeutic effects against food allergy. These findings preliminarily demonstrate the potential of probiotics to regulate food allergy. However, the results of clinical trials of probiotics in patients with food allergy have been controversial. Although most studies have shown that probiotics help prevent food allergy, some of the experimental results of these studies are invalid, and the results have been inconsistent and unstable. In addition, most studies of probiotics have focused on the bacteria themselves, rather than host–bacteria interactions, and the mechanism of action of probiotics on food allergies remains unclear. Furthermore, most data on probiotics for the treatment of food allergy are based on animal experiments. The small number of clinical studies in this field has been limited to observational studies, and there are few studies on the mechanism of probiotics in the treatment of food allergy. Moreover, medical studies suggest that, although uncommon, the use of probiotics may have adverse effects such as the spread of inappropriate resistance genes in the gut microbial populations, virulence factors in probiotic strains, transfer to tissues and blood, inflammation reactions, and infections. Therefore, further studies are needed to explore the role and mechanism of intestinal microbes in food allergy, to select more effective strains for treatment and prevention, and promote the establishment of new methods for regulating food allergy based on the gut microecology.

## 7. Conclusions

Food allergy has become a food safety and public health problem globally, and as such, it has garnered increasing research attention. In recent years, numerous animal and human studies have shown that the intestinal microbiota is strongly related to food allergy. Briefly, the intestinal microbiota composition of people with food allergy differs significantly from that of healthy people, and these differences are more evident in infants and children. Interactions between the microorganisms and microbial metabolites (SCFAs, tryptophan, and bile-acid) and the host immune response are probably the mechanism by which the microorganisms influence food allergy. Studies have shown that SCFAs modulate food allergy by activating GPRs, inhibiting HDAC activity, and strengthening the protection of the intestinal epithelium barrier. The application of probiotics to correct intestinal microbiota imbalances and regulate food allergy has become a research hotspot, and the combination of probiotics with other dietary interventions has been shown to have significant effects on preventing and treating disease. However, the mechanisms by which the intestinal microbiota regulates food allergy and the efficacy of probiotics are still in the preliminary exploration stage, and there are no clear and specific conclusions. In addition, the efficacy of probiotics also depends on the composition and activity of the host intestinal flora, as well as the derived metabolites and by-products of probiotics synthesis. Thus, further investigation is warranted in this field. With advances in the study of intestinal microbiota and its involvement in food allergy, we will have a greater understanding of food allergy, and it is expected that we will discover new strategies for the prevention and treatment of food allergy.

## Figures and Tables

**Figure 1 foods-11-02913-f001:**
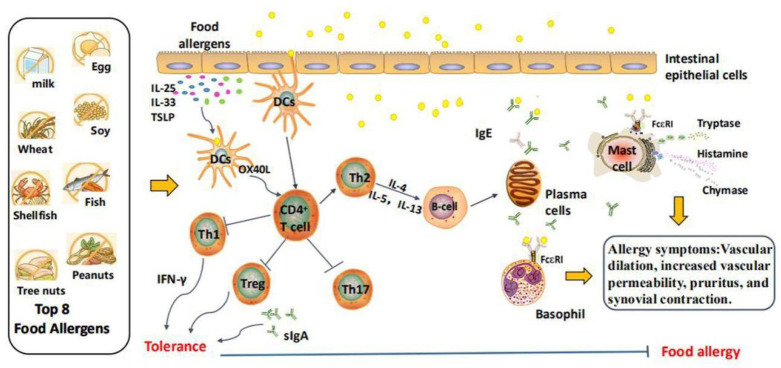
The possible mechanism of food allergy. Food allergens are digested into peptides in the gut through the epithelial barrier, which are captured by DC_S_ and presented to naive CD4^+^ T cells. Naive CD4^+^ T cells then differentiate into the Th-2 phenotype, producing IL-4, IL-5, and IL-13. Among them, IL-4 and IL-13 induce B cells to produce allergen-specific IgE antibodies, which bind to FcεRI receptors on the surface of basophils and mast cells, triggering degranulation and the release of chemical mediators such as histamine. TSLP: thymic stromal lymphopoietin; I-: inhibition.

**Figure 2 foods-11-02913-f002:**
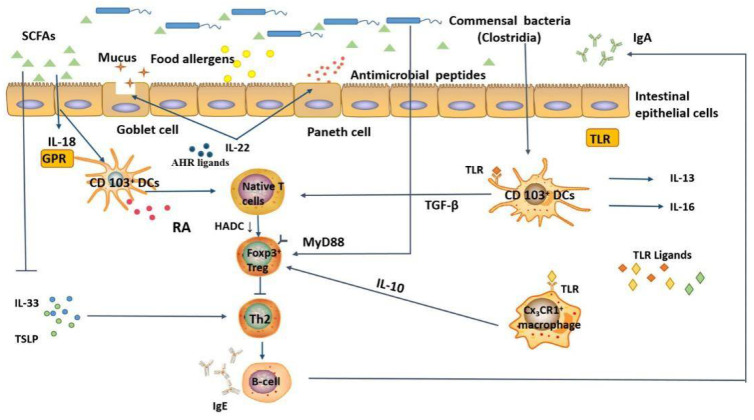
Potential mechanisms of intestinal microbiota and its metabolites in regulating food allergy, (1) regulation of allergen uptake and (2) regulation of signaling molecules involved in immune responses. Under normal circumstances, strengthening the intestinal mucosal barrier driven by commensal bacteria can reduce the entry of food allergens into the systemic circulation, thereby alleviating the symptoms of food allergy. When the balance between sensitization and oral tolerance is disrupted, signals (such as TLR, AHR) can be received by intestinal epithelial cells and DCs, enhancing the epithelial barrier, promoting Foxp3+ Treg cell induction, and preventing the generation of food-specific Th2 effector populations. SCFAs, short-chain fatty acids; TLR, toll-like receptor; GPR, G-protein coupled receptor; HDAC, histone deacetylase; AHR, aryl hydrocarbon receptor.

**Table 1 foods-11-02913-t001:** Correlation of gut microbiota in human subjects’ association of food allergy (↓: decrease; ↑: increase; —: no change).

Allergens Involved	Study Population	Age	Changes of Gut Microbiota in Food Allergy	Reference
Milk, eggs, wheat, nut, peanuts, fish, shrimp, soybeans	166 infants	0 to 1 year	↑ *Enterobacteriaceae/Bacteroidaceae ratio* ↓ *Ruminococcaceae*	[36]
Milk, eggs, wheat, nut, peanuts, fish, shrimp, soybeans	49 healthy infants 38 infants with food allergy	1 week to 12 months of age	↑ *Bifidobacterium* spp. — *Lactobacillus* spp. and *Clostridium perfringens*	[37]
Milk, eggs, wheat, nut, peanuts, fish, shrimp, soybean	34 infants with food allergy 45 healthy controls	2 to 11 years	↑ *Firmicutes and Fusobacteria* ↓ *Bacteroidetes, Proteobacteria and Actinobacteria* ↑ *Clostridium sensustricto* and *Anaerobacter* ↓ *Bacteroides* and *Clostridium XVIII*	[38]
Egg white, cow’s milk, wheat, peanut, soybean, gluten	23 children with food allergy 22 healthy children	6 to 24 months	↑ *Sphingomonas, Sutterella, Bifidobacterium*, *Collinsella*, *Clostridium sensustricto*, *Clostridium IV*, *Enterococcus, Lactobacillus*, *Roseburia*, *Faecalibacterium, Ruminococcus*, *Subdoligranulum,* and *Akkermansia*, ↓ *Bacteroides*, *Parabacteroides*, *Prevotella*, *Alistipes*, *Streptococcus,* and *Veillonella*	[39]
Egg, soybean, sesame, milk, shrimp, crab, peanut, wheat	4 children with food allergy 4 healthy children	18 months to 6 years	↓ *Dorea* and *Akkermansia* ↑ *Lachnospira*, *Veillonella, and Sutterella*	[40]
Egg	141 children with egg allergy	3 to 16 months	*Lachnospiraceae*, *Streptococcaceae*, and *Leuconostocaceae* families were differentially abundant in children with egg allergy	[42]
Cow milk	226 children with milk allergy	3 to 16 months	*Clostridia* and *Firmicutes* could be studied as probiotic candidates for milk allergy therapy	[43]
Peanuts, tree nuts, shellfish	1879 participants was 81.5%, ranging from 2.5% for peanuts to 40.5% for seasonal.	mean age, 45.5 years; 46.9% male	higher *Bacteroidales* and reduced *Clostridiales* taxa in nut and seasonal allergies	[44]
Egg, crab, shrimp	256 children with food allergy	4 to 12 years	*Bifidobacterium lactis* can effectively alleviate allergic reactions on food-specific IgE of food in children	[45]

**Table 2 foods-11-02913-t002:** Summary of proposed mechanistic actions of the effect that the gut microbial metabolites SCFAs on alleviating the food allergy.

SCFAs	Pathway	Proposed Mechanism	Reference
Acetate	GPR43	Promote gut IgA responses	[47]
Propionate	GPR41	Damage Th2 effector cells	[48]
Butyrate	GPR109a	Induction of Treg cells and IL-18 in the immune tolerance of food allergens	[49]
Acetate, propionate, butyrate	HDAC	Promote the differentiation of T cells into Th1, Th17 and Treg cells	[50]
Valerate	HDAC	Increase the production of anti-inflammatory factor IL-10	[51]
Acetate	HDAC9	Accelerate the production of Treg cells to alleviate allergic diseases	[52]

SCFAs: short-chain fatty acids, GPRs: G-protein coupled receptors, HDAC: histone deacetylase.

**Table 3 foods-11-02913-t003:** Summary of proposed mechanistic actions of the effect that the probiotics has on food allergy.

Subject	Source of Probiotics	Allergens Source	Therapies and Supplements	Effects and Mechanism of Action	Reference
human	*Lacticaseibacillus rhamnosus* GG (LGG)	Cow milk	Children with IgE-mediated cow milk allergy were randomly allocated to the LGG and followed for 36 months.	Reduced the incidence of the other allergic manifestations and hastens the development of oral tolerance in children with IgE-mediated cow’s milk allergy by a favorable modulation of gut microbiota and epigenetic mechanisms.	[69]
	LGG	Cow milk	Infant with IgE-mediated cow milk allergy were treated with extensively hydrolyzed casein formula either with or without supplementation with LGG (at 4.5 × 10^7^–8.5 × 10^7^ CFU/g) for 6 months.	Promoted tolerance in infants with cow’s milk allergy by influencing the strain-level bacterial community structure of the infant gut.	[70]
	*Lactobacillus helveticus* and *Streptococcus thermophilus*	α-lactalbumin and β-lactoglobul	Fermentation	The fermentation with lactic acid bacteria is an effective way to reduce whey proteins antigenicity.	[71]
	*Streptococcus thermophilus* and *Lactobacillus delbrueckii* subsp. *bulgaricus*	α-lactalbumin and β-lactoglobul		Lactobacillus fermented milk can not only interrupt allergen epitopes and produce some bioactive peptides, but also regulate the immune system.	[72]
	*Lactobacillus alimentarius*, *Lactobacillus brevis*, *Lactobacillus sanfranciscensis* and *Lactobacillus hilgardii*	Wheat allergens		Lactic acid bacteria fermentation was more conducive to the degradation of IgE epitopes of wheat protein.	[73]
	*Probiotic VSL#3 (Streptococcus thermophilus*, *Lactobacillus plantarum*, L. *acidophilus*, L. *casei*, L. *delbrueckii* spp. *bulgaricus*, *Bifidobacterium breve*, *B. longum* and *B. infantis)*	Wheat allergens		Probiotic VSL#3 can reduce the sensitizing proteins that cannot be degraded by pepsin and trypsin and reduce the potential risk of wheat protein to a certain extent.	[74]
animal	*Bifidobacterium*	Ovalbumin	Mice were orally administered with 200 mL/mouse of normal saline containing 10^8^ CFU/mL for 2 weeks.	Probiotics ameliorated allergic symptoms, including reducing OVA specific-IgE, and -IgG1 levels in the serum, Th2 cytokines release in spleen, and occurrence of diarrhea. Moreover, 16S rRNA analysis showed that the probiotics-mediated protection was conferred by an enrichment of *Coprococcus* and *Rikenella*.	[75]
	*Bifidobacterium infantis* 14.518 (Binf)	Shrimp-tropomyosin (TM)	Mice were daily administered with 500 µL/mouse Binf (2 × 10^9^ CFU/mL) for 3 weeks	Binf promotes the induction of Tregs and balance Th2/Treg for suppressing Th2 responses in TM-sensitized mice.	[76]
	*Lacticaseibacillus rhamnosus*, *Lactobacillus salivarius,* and *Bifidobacterium longum* subsp. *infantis*	Cow milk	10^9^ CFU/g for 6 weeks in mice	Lactobacillus salivarius strain blocked Th1 and Th2 responses, while the Bifidobacterium longum subsp. infantis strain induced a pro-Th1 profile and the *Lacticaseibacillus rhamnosus* strain induced pro-Th1 and regulatory response.	[77]
animal	*Lacticaseibacillus rhamnosus* and *Lacticaseibacillus casei*	Birch pollen	HEK293 cells of mice were stimulated with the formalin-inactivated *Lacticaseibacillus rhamnosus* and *Lacticaseibacillus* *casei* or their equal-part mixture at a concentration of 10^7^ CFU/mL for 20 h	Colonization with *lactobacilli* mixture inhibited the development of allergic immune responses through induction of regulatory cytokine TGF-β and can be thus exploited for alleviation of pollen allergies.	[78]
	LGG	Ovalbumin	Mice were given LGG for consecutive 22 days (low-dose LGG group, 1 × 10^8^ CFU/mL, 200 μL/d and high-dose LGG group, 1 × 10^9^ CFU/mL, 200 μL/d	*Lacticaseibacillus rhamnosus* GG can remarkably improve the symptoms of ovalbumin-induced food allergy probably by decreasing IL-4/IFN-γ ratio.	[79]
	*La-Dahi* *(Lactobacillus acidophilus LaVK2* and *Bifidobacterium bifidum BbVK3)*	Whey proteins	2 × 10^9^ CFU/g for 1 week in mice	Probiotic Dahi skewed Th2-specific immune response towards Th1-specific response and suppressed IgE in serum	[80]
	*Bifidobacterium breve* M-16V	Ovalbumin	1 × 10^7^ CFU/g formula per mouse for 40 days	Bifidobacterium breve M-16V may alter the gut microbiota to alleviate the allergy symptoms by IL-33/tumorigenicity 2 signaling	[81]
	*Lacticaseibacillus casei* BL23	Cow milk	Mice were intragastrically administered with *Lacticaseibacillus* casei BL23 (~1.5 × 10^8^ CFU/mouse/administration) for 5 consecutive days.	Intragastric administration of *Lacticaseibacillus casei* BL23 to mice induces local and systemic Foxp^3+^ RORγt^+^ type 3 Treg cells, the specific transcription factor of Th17 cells.	[82]

## Data Availability

The data presented in this study are available on request from the corresponding author.

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
