# Peer review of "Research Progress on the Correlation between the Intestinal Microbiota and Food Allergy"

_foods, 2022, doi:10.3390/foods11182913_

Round 1

Reviewer 1 Report

The manuscript (Research progress on the correlation between the intestinal microbiota and food allergy) has some good ideas, but it needs to add some correctives and modifications.

1-The summary in the manuscript is weak and needs to be rewritten again.

2- The writing references style is not matching with Foods Journal style.

3-Some paragraphs in the manuscript need to add references, and I suggested adding some modern references such as page 4 line 133, (Al-Sahlany, S. T. G., Khassaf, W. H., Niamah, A. K., & Abd Al-Manhel, A. J. (2022). Date juice addition to bio-yogurt: The effects on physicochemical and microbiological properties during storage, as well as blood parameters in vivo. Journal of the Saudi Society of Agricultural Sciences.‏ https://doi.org/10.1016/j.jssas.2022.06.005)

Page 4 Line 137, Martens, E. C., Neumann, M., & Desai, M. S. (2018). Interactions of commensal and pathogenic microorganisms with the intestinal mucosal barrier. Nature Reviews Microbiology, 16(8), 457-470.‏

Page 4 line 154, (Rotkiewicz, T. A. D. E. U. S. Z., Rotkiewicz, Z. O. F. I. A., Depta, A. N. D. R. Z. E. J., & Kander, M. A. L. G. O. R. Z. A. T. A. (2001). Effect of Lactobacillus acidophilus and Bifidobacterium sp. on the course of Cryptosporidium parvum invasion in new-born piglets. BULLETIN-VETERINARY INSTITUTE IN PULAWY, 45(2), 187-196.‏)

4- Page 6 Table 1 , The arrows in Table 1 must be explained below the table and indicate all the symbols present.

5-  Page 6 Table 1 , The names of the bacteria must be written in italics throughout the manuscript.

6-Page 8 line 337, What is this symbol mean?

7-Page 11 Table 3 , The table needs to add modern reference  such as 

1- Al-Sahlany, S.T.G. and Niamah, A.K. (2022), "Bacterial viability, antioxidant stability, antimutagenicity and sensory properties of onion types fermentation by using probiotic starter during storage", Nutrition & Food Science, Vol. 52 No. 6, pp. 901-916. https://doi.org/10.1108/NFS-07-2021-0204

2-Wu, Z., Mehrabi Nasab, E., Arora, P., & Athari, S. S. (2022). Study effect of probiotics and prebiotics on treatment of OVA-LPS-induced of allergic asthma inflammation and pneumonia by regulating the TLR4/NF-kB signaling pathway. Journal of Translational Medicine, 20(1), 1-14.

8-Page 11 Table 3, The use of the modern nomenclature for lactic acid bacteria such as  Lactobacillus rhamnosus correct to  Lacticaseibacillus rhamnosus and Lactobacillus casei correct to Lacticaseibacillus casei 

Author Response

On behalf of all the co-authors, many thanks to you for offering us an opportunity to revise the manuscript, and we also appreciate you and reviewers very much for your positive and constructive comments and suggestions on the manuscript entitled “Research progress on the correlation between the intestinal microbiota and food allergy” (foods-1879455).

We have read your comments carefully and have tried our best to revise our manuscript according to the comments, and the revision was marked in red in the manuscript.

Yours sincerely,  
Hui Yang

Rina Wu

[email protected]

Special thanks for your kind comments.

  1. The summary in the manuscript is weak and needs to be rewritten again.

√Thank you for pointing it out, and we have rewritten this part according to your comment. (Line 9-14)

  1. The writing references style is not matching with Foods Journal style.  

√Thank you for pointing it out, and we have made correction according to your comment.

  1. Some paragraphs in the manuscript need to add references, and I suggested adding some modern references such as page 4 line 133, (Al-Sahlany, S. T. G., Khassaf, W. H., Niamah, A. K., & Abd Al-Manhel, A. J. (2022). Date juice addition to bio-yogurt: The effects on physicochemical and microbiological properties during storage, as well as blood parameters in vivo. Journal of the Saudi Society of Agricultural Sciences.‏ https://doi.org/10.1016/j.jssas.2022.06.005).

√Thank you for pointing it out, and we have made correction according to your comment. (reference 13 )

Page 4 Line 137, Martens, E. C., Neumann, M., & Desai, M. S. (2018). Interactions of commensal and pathogenic microorganisms with the intestinal mucosal barrier. Nature Reviews Microbiology, 16(8), 457-470.‏

√Thank you for pointing it out, and we have made correction according to your comment. (reference 14 )

Page 4 line 154, (Rotkiewicz, T. A. D. E. U. S. Z., Rotkiewicz, Z. O. F. I. A., Depta, A. N. D. R. Z. E. J., & Kander, M. A. L. G. O. R. Z. A. T. A. (2001). Effect of Lactobacillus acidophilus and Bifidobacterium sp. on the course of Cryptosporidium parvum invasion in new-born piglets. BULLETIN-VETERINARY INSTITUTE IN PULAWY, 45(2), 187-196.‏)

√Thank you for pointing it out, and we have made correction according to your comment. (reference 17 )

  1. Page 6 Table 1 , The arrows in Table 1 must be explained below the table and indicate all the symbols present.

√Thank you for pointing it out, and we have made correction according to your comment. (Line 271)

  1. Page 6 Table 1 , The names of the bacteria must be written in italics throughout the manuscript.

√Thank you for pointing it out, and we have made correction according to your comment. (Table 1and Table 3)

  1. Page 8 line 337, What is this symbol mean?

√Thank you for pointing it out, and we have made correction according to your comment. (Line 342)

Table 1. Changes of gut microbiota in food allergy (↓: decrease; ↑:increase;-:no change).

  1. Page 11 Table 3 , The table needs to add modern reference  such as 
  • Al-Sahlany, S.T.G. and Niamah, A.K. (2022), "Bacterial viability, antioxidant stability, antimutagenicity and sensory properties of onion types fermentation by using probiotic starter during storage", Nutrition & Food Science, Vol. 52 No. 6, pp. 901-916. https://doi.org/10.1108/NFS-07-2021-0204

√Thank you for pointing it out, and we have made correction according to your comment. (reference 68 )

  • Wu, Z., Mehrabi Nasab, E., Arora, P., & Athari, S. S. (2022). Study effect of probiotics and prebiotics on treatment of OVA-LPS-induced of allergic asthma inflammation and pneumonia by regulating the TLR4/NF-kB signaling pathway. Journal of Translational Medicine, 20(1), 1-14.

√Thank you for pointing it out, and we have made correction according to your comment. (Line 504-505, reference 87 )

8-Page 11 Table 3, The use of the modern nomenclature for lactic acid bacteria such as  Lactobacillus rhamnosus correct to  Lacticaseibacillus rhamnosus and Lactobacillus casei correct to Lacticaseibacillus casei 

√Thank you for pointing it out, and we have made correction according to your comment. (Table 3)

Reviewer 2 Report

REVIEWER COMMENTS

COMMENTS FOR THE AUTHOR(S)

The manuscript titled “Research progress on the correlation between the intestinal microbiota and food allergy” is a paper focused on an extremely current and important area of ​​contemporary allergology. This is another review article on this subject, but it stands out from the others with a very broad approach to the problem. The content of the work covers both selected information on the mechanisms of development of food allergy, the influence of the microbiome on GALT, including issues related to epigenetics and metabolomics, influence of selected patterns of diet on microbiota as well as topics related to probiotic therapy.

The manuscript is nicely structured, quite interesting and takes into account the wide spectrum of studies results on the correlation between the intestinal microbiota and food allergy.

In my opinion, I see the need for a few changes that could help deliver the ideas/views.

Major comments:

1.  Although the authors try to summarize the information contained therein, there is no attempt to identify "good" from "bad" bacteria (i.e. summary) on the basis of numerous studies carried out.

2.  In the last parts of probiotic therapy, attention is drawn to the lack of indication of the criteria for the selection of the literature. There are many studies on probiotic therapy in allergology, including those conducted on humans (the authors refer equally to various studies based on experiments on animals and in vitro).
Table 3 - division into human, animal studies and fermentation is advisable.

3.  Line 100,101 - “Despite high exposure to food allergens, only a few people experience adverse immunological reactions. This means that the immune system does not generate a response to allergens via the oral route”.- I agree with the first sentence, but the second sentence needs a correction.

Minor comments:

1.  Fig. 2 - please describe the legend better

2.  Line 174 - instead of “Influence of dietary intakes in intestinal microbiota composition” it seems better “Influence of dietary intakes on intestinal microbiota composition”

Author Response

On behalf of all the co-authors, many thanks to you for offering us an opportunity to revise the manuscript, and we also appreciate you and reviewers very much for your positive and constructive comments and suggestions on the manuscript entitled “Research progress on the correlation between the intestinal microbiota and food allergy” (foods-1879455).

We have read your comments carefully and have tried our best to revise our manuscript according to the comments, and the revision was marked in red in the manuscript.

Yours sincerely,  
Hui Yang

Rina Wu

[email protected]

Special thanks for your kind comments.

  1. Although the authors try to summarize the information contained therein, there is no attempt to identify "good" from "bad" bacteria (i.e. summary) on the basis of numerous studies carried out.

√Thanks for your precious suggestions, you are right.

So far, researches on the characteristics of the intestinal microbiota in patients with food allergy are still in its infancy, and no specific bacterial taxa has been identified that may be associated with the occurrence of food allergy (Bifidobacterium and Lactobacillus has been shown to inhibit allergic inflammation in the lungs). Also, a main limitation of all these researches is that the number of patients with sensitization or food allergy is small, such that statistical analyses of the effects of potential confounding variables, such as delivery modes, breastfeeding, diet, antibiotic intake, and pets, have not been possible. In the future, I will continue to pay attention to this aspect of research.

  1. Yours include references outside this range.In the last parts of probiotic therapy, attention is drawn to the lack of indication of the criteria for the selection of the literature. There are many studies on probiotic therapy in allergology, including those conducted on humans (the authors refer equally to various studies based on experiments on animals and in vitro). 

√Thanks for your precious suggestions, you are right. There are many studies on probiotic therapy in allergology, including those conducted on humans. In this study, we focus on the studies of people with food allergy, such as, Capurso (2019) found that administration of Lacticaseibacillus rhamnosus GG or a Bifidobacterium complete hydrolysis formula significantly improves the condition of skin and ameliorates inflammatory indicators in infants with atopic eczema. Jerzynska et al. (2016) confirmed clinical and immunologic effects of probiotic and vitamin D supplementation for sublingual immunotherapy, for example, in children with allergic rhinitis, where probiotic supplementation demonstrated better clinical and immunotherapeutic effects. Birch pollen allergy has been associated with changes in the fecal microbiota composition, and a probiotic combination of Lactobacillus acidophilus NCFMTM (ATCC 700396) and Bifidobacterium lactis Bl-04 (ATCC SD5219) has been shown to prevent the pollen-induced eosinophil infiltration into the nasal mucosa and promote a tendency to reduce nasal symptoms. However, the long-term efficacy of probiotics in the prevention and treatment of allergic disease needs to be further evaluated. Plummer et al. (2020) found that postnatal administration of probiotics had no effect on the incidence of allergic disease or atopic sensitization in preterm children during the first 2 years. Moreover, West et al. (2013) reported that Lactobacillus paracasei ssp paracasei F19 had no effect on any diagnosed allergic disease, airway inflammation, or IgE sensitization. It is worth noting that the results vary from study to study due to multiple factors (hose, probiotic supplement and environment). (Line 510-528)

Table 3 - division into human, animal studies and fermentation is advisable.

√Thank you for pointing it out, we have made a correction according to your comment (Table 3).

  1. Line 100,101 - “Despite high exposure to food allergens, only a few people experience adverse immunological reactions. This means that the immune system does not generate a response to allergens via the oral route”.- I agree with the first sentence, but the second sentence needs a correction.

√Thank you for pointing it out, we have made a correction according to your comment (Line 104-105).

Despite high exposure to food allergens, only a few people experience adverse immunological reactions. The body's immune failure to respond to food allergens is called immune tolerance.

  1. Fig. 2 - please describe the legend better

√Thank you for pointing it out, we have made a correction according to your comment (Fig.2). 

Potential mechanisms of intestinal microbiota and its metabolites in regulating food allergy, (1) regulation of allergens uptake and (2) regulation of signalling molecules involved in immune responses. Under normal circumstances, strengthening the intestinal mucosal barrier driven by commensal bacteria can reduce the entry of food allergens into the systemic circulation, thereby alleviating the symptoms of food allergy. When the balance between sensitization and oral tolerance is disrupted, signalings (such as TLR, AHR) can be received by intestinal epithelial cells and DCs, enhancing the epithelial barrier, promoting Foxp3+ Treg cell induction, and preventing the generation of food-specific Th2 effector populations. SCFAs, short-chain fatty acids; TLR, toll-like receptor; GPR, G-protein coupled receptor; HDAC, histone deacetylase; AHR, aryl hydrocarbon receptor.

       5. Line 174 - instead of “Influence of dietary intakes in intestinal microbiota composition”it seems better “Influence of dietary intakes on intestinal microbiota composition”.

√Thank you for pointing it out, we have made a correction according to your comment (Line 177).

Round 2

Reviewer 1 Report

Dear Editors, 

The authors made all the necessary changes to improve the manuscript, and now I recommend it for publication in its current form.

Author Response

Special thanks for your kind comments. We have now worked on both language and readability and have also involved native English speakers for language corrections. We really hope that the language level and logic have been substantially improved.